# Sex-Specific Effects of the Genetic Variant rs10487505 Upstream of *leptin* in the Development of Obesity

**DOI:** 10.3390/genes14020378

**Published:** 2023-01-31

**Authors:** Janine Molder, Esther Guiu-Jurado, Yusef Moulla, Christine Stroh, Arne Dietrich, Michael R. Schön, Daniel Gärtner, Tobias Lohmann, Miriam Dressler, Michael Stumvoll, Peter Kovacs, Matthias Blüher, Nora Klöting

**Affiliations:** 1Medical Department III—Endocrinology, Nephrology, Rheumatology, University of Leipzig Medical Center, 04103 Leipzig, Germany; 2Deutsches Zentrum für Diabetesforschung e.V., 85764 Neuherberg, Germany; 3Clinic for Visceral, Transplantation and Thorax and Vascular Surgery, University Hospital Leipzig, 04103 Leipzig, Germany; 4Department of General, Abdominal and Pediatric Surgery, Municipal Hospital, 07548 Gera, Germany; 5Städtisches Klinikum Karlsruhe, Clinic of Visceral Surgery, 76133 Karlsruhe, Germany; 6Municipal Clinic Dresden-Neustadt, 01129 Dresden, Germany; 7Helmholtz Institute for Metabolic, Obesity and Vascular Research (HI-MAG) of the Helmholtz Zentrum München at the University of Leipzig and University Hospital Leipzig, 04103 Leipzig, Germany

**Keywords:** leptin, rs10487505, obesity, adipose tissue

## Abstract

The SNP rs10487505 in the promotor region of the *leptin* gene was reported to be associated with decreased circulating leptin and increased body mass index (BMI). However, the phenotypic outcomes affected by rs10487505 in the leptin regulatory pathway have not been systematically studied. Therefore, the aim of this study was to elucidate the influence of rs10487505 on *leptin* mRNA expression and obesity-related parameters. We genotyped rs10487505 in DNA samples from 1665 patients with obesity and lean controls and measured *leptin* gene expression in paired samples of adipose tissue (AT, N = 310), as well as circulating leptin levels. We confirm the leptin-lowering effect of rs10487505 in women. In contrast to the previously reported data from population-based studies, in this mainly obese cohort, we describe a lower mean BMI in women carrying the C allele of rs10487505. However, no association of rs10487505 with AT *leptin* mRNA expression was found. Our data suggest that reduced circulating leptin levels are not a result of the direct silencing of *leptin* mRNA expression. Furthermore, leptin reduction by rs10487505 does not associate with BMI in a linear manner. Instead, the decreasing effect on BMI might be dependent on the severity of obesity.

## 1. Introduction

Obesity has become a global epidemic with a huge individual and socioeconomic impact. It acts as a risk factor for a wide spectrum of diseases such as type 2 diabetes mellitus (T2D), non-alcoholic fatty liver disease (NAFLD), cardiovascular diseases such as arterial hypotension, ischemic heart attack, ischemic stroke and thromboembolism, osteoarthritis, and certain types of cancer [1,2].

In 1994, Zhang et al. identified *leptin* as the gene product of the *obese* gene, a gene locus known to be associated with obesity since the 1950s [3,4]. Leptin is a hormone mainly secreted by adipose tissue (AT) [5]. It affects gene expression in specific neuronal clusters, predominantly in the hypothalamus [6,7,8]. Molecular effectors regulate appetite/satiety, energy metabolism and secretion of hormones of the thyrotropic and gonadotropic axis. Congenital leptin deficiency leads to severe dysregulation of satiety and food intake resulting in obesity [9]. In contrast, obese individuals exhibit elevated leptin levels that have been attributed to leptin resistance [10]. *Leptin* (*LEP)* gene expression is typically higher in women than in men [11]. This is due to a differential expression of *LEP* mRNA in the two large fat depots of the body. Subcutaneous adipose tissue (SCAT) shows a higher *LEP* gene expression than visceral adipose tissue (VAT) [7]. Young women tend to accumulate SCAT predominantly, while men show a higher propensity to accumulate VAT [12]. The regulation of *leptin* gene expression is the subject of recent studies. In 2016, a genome-wide association study (GWAS) identified the single nucleotide polymorphism (SNP) rs10487505 in the proximal promotor region of the *leptin* gene, which was associated with lower leptin levels and increased body mass index (BMI) [13]. This variant is of great interest because it is located in a long non-coding RNA (lncOb). The equivalent of this lncOb correlated positively with *leptin* gene expression in mice [14]. Taken together, these recent findings propose that the SNP rs10487505 may have a role in the development of obesity and obesity-related comorbidities.

We, therefore, tested the hypothesis that rs10487505 is related to obesity, fat distribution and parameters of AT function. We genotyped the rs10487505 in DNA samples from 1665 individuals predominantly with obesity and measured *leptin* mRNA expression in paired samples of AT as well as circulating leptin levels. In a linear regression model, we analyzed if rs10487505 influences circulating leptin levels as well as obesity-associated parameters in this cohort and if the SNP is an expression quantitative trait locus (eQTL) for *leptin*.

## 2. Materials and Methods

### 2.1. Subjects

The cohort consists of 1665 metabolically well-characterized individuals (Table 1) of the Leipzig Obesity BioBank recruited at bariatric surgery centers in Leipzig, Karlsruhe, Dresden and Gera (all in Germany). All patients included in this study had a stable weight, defined as fluctuations of < 2% of body weight for at least 3 months before surgery. In addition, clinical and anthropometric parameters were routinely recorded as described previously [15]. According to ADA criteria [16], 39.0% of the patients were diagnosed with T2D, 47.0 % had normal glucose tolerance (NGT), and 11.6% exhibited a prediabetic state such as impaired glucose tolerance (IGT), impaired fasting glucose (IFG) or insulin resistance (IR). Moreover, patients diagnosed with acute or chronic hepatic, inflammatory, infectious, or neoplastic diseases were excluded from the study. The study was approved by the ethics committee of the University of Leipzig (Approval numbers: 159-12-21052012 and 017-12-23012012). The study design follows the Declaration of Helsinki, and all subjects in this study gave written informed consent to participate in medical research.

### 2.2. Leptin Gene Expression Analyses in AT

Adipose tissue samples were obtained in pairs of abdominal omental (visceral, VAT) and subcutaneous (SCAT) AT from 585 Caucasian men (N = 156) and women (N = 429) during elective laparoscopic bariatric surgery as described previously [17]. The mean age of this subgroup was 47.51 ± 12.41 years, and the mean BMI was 47.71 ± 8.80 kg/m^2^. After surgery, AT samples were immediately frozen in liquid nitrogen and stored at −80°C. RNA extraction from AT was performed using RNeasy Lipid Tissue Mini Kit (Qiagen, Hilden, Germany), and quantitative (q) PCR was carried out as described [17]. Real-time qPCR was performed using TaqMan Assay predesigned by Applied Biosystems (Foster City, CA, USA) for the detection of human *leptin* (Hs00174877_m1) and *Eukaryotic 18S rRNA* (18S) (Hs99999901_s1) gene expression in VAT and SCAT. All reactions were performed in 96-well plates using the QuantStudioTM 6 Flex System Fast Real-Time PCR system. Relative *leptin* mRNA expression was obtained after normalization to the endogenous control gene *18S* using the formula 2^–∆∆Ct^.

### 2.3. Measurement of Leptin Serum Concentrations

Leptin serum concentrations were analyzed in duplicate using enzyme-linked immunosorbent assay (ELISA) according to the manufacturer’s instructions (Leptin Human ELISA, Mediagnost, Reutlingen, Germany) in 626 patients. The mean age of these subjects was 48.13 ± 12.84 years, and the mean BMI was 49.21 ± 12.09 kg/m^2^. Leptin assay sensitivity was <0.25 pg/mL, and the inter-assay and intra-assay coefficients of variation were less than 15%.

### 2.4. Genotyping

Genotyping of the SNPs rs10487505 was done in 1665 individuals according to the manufacturer’s protocol using the SNP genotyping probe C_1331246_10 (Thermo Fisher Scientific, Waltham, Massachusetts, USA) and the ABI PRISM 7500 Sequence Detecting System (Life Technologies, Foster City, CA, USA). For quality control, randomly selected samples (5%) have been re-genotyped for rs10487505 SNP. All genotypes matched the initially designated genotypes. Genotype distribution was according to Hardy–Weinberg Equilibrium, all presenting *p* > 0.05. C is the minor allele.

### 2.5. Statistical Analysis

Statistical analysis was carried out using the IBM SPSS Statistic 27 software (IBM Corp., Armok, NY, USA). Prior to statistical analysis, variables were tested for normal distribution using Kolmogorov–Smirnov test. To achieve normal distribution, the variables included in this study were logarithmically transformed. Differences in mRNA expression between fat depots were performed using the paired Student’s *t*-test. An unpaired *t*-test was used to assess differences in AT gene expression among different phenotypes. The association of the SNP and diabetes status was conducted using logistic regression analyses. A linear regression model was used to analyze the relationship between genetic variants/gene expression and quantitative metabolic traits. Pearson´s correlation analyses were conducted using two-way bivariate correlations. The additive model (with genotypes coded to 0, 1 and 2) was used, and, if not otherwise indicated, all *p*-values were adjusted for age, sex and BMI. Significant evidence for nominal association was assumed at a two-sided *p*-value ≤ 0.05, and no correction for multiple hypothesis testing was performed.

## 3. Results

### 3.1. Baseline Characteristics of the Subjects

The main characteristics of the cohort studied, including anthropometric and biochemical (liver enzymes, glucose and lipid metabolism) parameters, are shown in Table 1. Our cohort of 1665 individuals (1113 women and 522 men) was classified according to BMI. Mean age was significantly increased in individuals with BMI < 30 kg/m^2^ compared to those with BMI ≥ 30 kg/m^2^. In terms of anthropometric measurements (weight, BMI, waist and hip circumference, waist-to-hip ratio (WHR) and body fat (%)), there were significant differences between groups, being increased in patients with BMI ≥ 30 kg/m^2^. Biochemical analyses indicate that parameters involved in glucose metabolism (fasting plasma glucose (FPG), fasting plasma insulin (FPI) and glycosylated hemoglobin (HbA1c)) were also significantly higher in patients with a higher BMI. With regard to lipid parameters, total cholesterol was significantly increased in patients with BMI ≥ 30 kg/m^2^, while high-density lipoprotein cholesterol (HDL-C) was higher in normal-weight and overweight patients. As far as transaminases are concerned, Table 1 shows that levels of alanine aminotransferase (ALAT) were significantly higher in patients with obesity than in normal-weight and overweight individuals. However, levels of γ-glutamyltransferase (gGT) and aspartate aminotransferase (ASAT) were not significantly different between groups. Finally, with regard to adipokines and inflammation parameters, interleukin-6 (IL-6) and leptin levels were significantly higher in patients with obesity than in normal-weight subjects, whereas adiponectin levels were significantly lower in individuals with obesity. But, C-related protein (CRP) levels were not significantly different between the groups studied.

### 3.2. Characteristics of Leptin Gene Expression in a Cohort with Obesity

*LEP* mRNA was significantly higher expressed in SCAT than in VAT (Figure 1a). Due to the sex-dependent gene expression of LEP, in all further analyses, we examined LEP mRNA in the two fat depots separated by sex. In both men and women, LEP was predominantly expressed in SCAT. Furthermore, LEP mRNA was significantly higher in women’s SCAT than in men, whereas no difference was observed between the LEP expression of women and men in VAT (Figure 1b).

We performed additional analyses comparing the subgroup of normal weight and overweight (BMI < 30 kg/m^2^) individuals with individuals of obesity (BMI ≥ 30 kg/m^2^). *LEP* mRNA levels were higher in patients with obesity in both fat depots regardless of sex (Figure 1c,d).

We further stratified patients according to their state of glucose tolerance. SCAT and VAT *LEP* mRNA expression were not significantly different across participants with normal or impaired glucose tolerance and T2D, independent of sex (Figure 1e,f).

### 3.3. rs10487505 Is Associated with Circulating Leptin in Women but not with LEP mRNA in SCAT

Associations of rs10487505 with circulating leptin, *LEP* mRNA and obesity-related parameters were assessed in a linear regression model. Results are summarized in Appendix A. In women, the rs10487505 C allele was significantly associated with lower circulating leptin levels (Figure 2; Appendix A). There were no associations of the SNP with *LEP* gene expression in both sexes and in both examined fat depots (Figure 2). Albeit not significantly, the *leptin* gene expression according to genotypes follows a similar pattern as circulating leptin, especially in SCAT of women (Figure 2).

In a mediation analysis, we tested whether the effect of rs10487505 on circulating leptin was mediated to some extent by *LEP* mRNA expression (Appendix A). We were able to confirm the direct effect of the SNP on circulating leptin in women in the model with *LEP* gene expression in VAT. However, the mRNA expression itself was not a mediator for this effect in both SCAT and VAT.

### 3.4. rs10487505 C-allele Is Associated with BMI and Metabolic Parameters

In women, the leptin-reducing C allele of rs10487505 was associated with lower BMI and fasting plasma insulin (FPI) in the additive inheritance mode (Figure 3). Mediation analysis did not show a significant mediating effect of *LEP* mRNA expression on BMI in either fat depot. However, *LEP* mRNA expression in VAT seems to have an effect on BMI without being influenced by the SNP (Appendix A).

Logistic regression analysis showed that the SNP does not influence the development of T2D, neither in the entire cohort nor only in women (Table 2).

In the other inheritance modes, rs10487505 showed associations with gGT and IL-6 in women with a reducing effect of the C allele. In men, the C allele was associated with lower mean FPI and body fat and higher adiponectin levels (Appendix A).

### 3.5. The BMI Decreasing Effect Is Specific to Postmenopausal Women

Since the effects of the SNP on circulating leptin and BMI were only found in women, we analyzed the correlation of the SNP with all parameters studied in pre (<40 years)- and postmenopausal (≥60 years) women. The leptin-lowering effect was detected in both age groups; however, the decreasing effect on BMI was only found in postmenopausal women (Appendix A).

## 4. Discussion

Obesity is a disease with an enormous impact on patients’ life quality. Understanding the mechanisms underlying obesity is therefore critical for targeted therapy and the prevention of its related comorbidities. A GWAS in 2016 revealed the SNP rs10487505, which was associated with decreased circulating leptin and increased BMI [13]. Therefore, this SNP is a plausible genetic variant that is involved in the development of obesity, although its effects are still unknown. A previous study indicated that rs10487505 was not associated with *LEP* mRNA expression [13], but further studies are inevitable to better understand the regulation of *LEP* gene expression. In this work, we examined whether rs10487505 was associated with *LEP* gene expression in AT in a cohort predominantly with obesity and tested its associations with several obesity-related parameters.

### 4.1. rs10487505 Is Associated with Circulating Leptin Levels, but the Association Is not Mediated by LEP mRNA Expression

We were able to confirm the leptin-lowering effect of the C allele in women. Maurano et al. reported that about 90% of disease- and trait-associated SNPs are located in noncoding regions of the genome [18]. This includes, for example, promoter sequences, 5′ and 3′ untranslated mRNA regions and lncRNAs [19]. In 2019, Dallner et al. found evidence that rs10487505 is located within the sequence of a lncRNA named lncOb. A homologous lncRNA in mice correlated positively with *Lep* gene expression and circulating leptin levels. In a small sample (n = 7), Dallner et al. found a correlation between this lncRNA and *LEP* mRNA expression in humans [14]. Unfortunately, the exact sequence of the of LncOb was not publically available, and its gene expression could not be measured in this study. However, we investigated whether rs10487505 interferes with leptin synthesis on the level of *LEP* mRNA expression. Neither did we find an association of the SNP with *LEP* expression levels, nor did we observe a mediating effect of *LEP* mRNA expression on circulating leptin levels in our mainly obese cohort. These findings are consistent with data from previous association analyses in population-based cohorts [13], suggesting that the SNP may not or not exclusively regulate leptin protein levels via direct silencing of the transcription of the *LEP* gene, for example, through direct interaction with a lncRNA. Regulatory mechanisms may occur further downstream in the process of protein synthesis, for example, through reduced translation frequency or secretion into the bloodstream or increased degradation of the leptin protein. Further studies are needed to uncover the effectors of rs10487505. An evaluation of topologically-associated domains could give a hint for potential interaction partners. Proteins and RNA binding in the region of the SNP could be identified with pull-down assays in human cultured adipocytes.

Since the leptin-reducing effect of the SNP also appears to be specific to women, we examined whether the effects of the SNP depend on the female hormonal status by comparing the association of the SNP with circulating leptin levels in a pre- and postmenopausal woman. The leptin-reducing effect of the SNP seems to be independent of estrogen levels since the respective associations were detected in both age groups.

### 4.2. rs10487505 C Allele Is Associated with Lower BMI in a Cohort with Obesity

In our study, we found a decreasing effect of the C allele (leptin-lowering allele) of rs10487505 on BMI in women that was not mediated by AT *LEP* gene expression. This contradicts the findings from the original GWAS performed in the cohort of the GIANT Consortium, which described an association with higher BMI (N = 221,677, *p* = 0.03) [13]. In contrast, to study cohorts commonly included in large-scale GWAS and their meta-analyses, our work was performed in a cohort with morbid obesity with an average BMI of 47.1 kg/m^2^ (women: 47.2 kg/m^2^; men = 46.8 kg/m^2^). Therefore, we suggest that rs10487505 does not influence the BMI in a linear manner. Instead, the leptin-lowering allele may have an increasing effect on BMI within a normal BMI range. In the BMI range of individuals with morbid obesity, its effect may instead be decreasing. Furthermore, in the pre- and postmenopausal age-stratified analysis, the increasing effect on BMI appeared to be a postmenopausal phenomenon. One hypothesis for the connection between leptin depletion by rs10487505 and its effect on BMI could be that, due to normal leptin sensitivity, normal-weight individuals react to leptin depletion by gaining weight. In contrast, in patients with morbid obesity, leptin depletion might act as a preventative factor for the development of leptin resistance, therefore preserving the physiological regulation of satiety and appetite to some extent. A prerequisite for the BMI-decreasing effect appears to be the postmenopausal age (although not age as a linear parameter) of women. Typically, a decrease in estrogen levels after menopause is associated with a higher risk of central obesity, insulin resistance and CVD [20]. Possibly, the more severe metabolic dysregulation in the postmenopausal subgroup further enhances the BMI-decreasing effect of the C allele.

### 4.3. rs10487505 Correlates with Obesity-Related and Inflammatory Parameters

In addition to its association with BMI, we also found associations of rs10487505 with other parameters of obesity and inflammation. In women, independent of obesity, the leptin-reducing allele increased height and decreased FPI, the cholestasis parameter gGT and the cytokine IL-6. In men, it correlated with lower body fat and FPI and higher adiponectin. Lower FPI is indicative of better insulin sensitivity [21]. High levels of adiponectin are associated with a reduced risk of T2D, atherosclerosis and less inflammation [22]. In general, these results suggest a novel, rather protective role of the leptin-lowering allele in individuals with obesity. This inference is supported by the fact that rs10487505 is not associated with the state of glucose intolerance in our study. Thus, rs10487505 could serve as a predictor for more or less severe progress of obesity and its comorbidities. In this context, it would also be interesting to investigate the association of the SNP with clinical complications like cardiovascular events or obesity-related cancer.

Our study has some limitations. First, we predominantly included patients undergoing bariatric surgery with obesity. This high BMI bias needs to be acknowledged. Consequently, our results may not reflect the effects of the SNP in the lower BMI range. Furthermore, typically the number of women undergoing bariatric surgery is significantly higher than that of men [23]. This discrepancy can also be observed in this study. The statistical power of the data assessed in women is higher than in men. On the other hand, we described a gender-specific effect of rs10487505 and a possible effect of menopausal status. Regrettably, gender and menopausal markers such as estradiol, estrone, testosterone, luteinizing hormone (LH) and follicle-stimulating hormone (FSH) were not measured in this study to better assess these subgroups. Further studies, including data regarding these hormones, are needed to gain further knowledge about the influence of gender and menopausal status on leptin and obesity through rs10487505. Another aspect to take into account is that association studies of rs10487505 with leptin have been performed almost exclusively in European cohorts [24]. Interestingly, MAF varies widely among populations of different ancestry, and BMI is also one of the anthropometric traits with large differences between large populations [25]. Therefore, it is possible that the impact of this genetic variant on leptin and obesity varies in these populations, so it is important to consider the ancestry of the cohort studied in future studies. Finally, to better understand the exact mechanisms of leptin regulation by rs10487505, it would be of interest to identify the interaction or direct effects of the SNP.

## 5. Conclusions

We assessed the effects of the SNP rs10487505 on obesity-related traits. We confirmed the leptin-lowering effect of the C allele, as reported previously [13]. In our study, *LEP* gene expression was not significantly affected by the SNP, suggesting that, as a result of the specific SNP genotype, leptin levels might not be exclusively regulated at the mRNA level. We also report that rs10487505 is associated with decreased BMI in a predominantly obese cohort (in contrast to a BMI-increasing effect in previous population-based studies) and is associated with an improvement in several metabolic parameters related to obesity and its comorbidities. Taken together, we suggest a model in which leptin depletion via rs10487505 enhances weight gain in normal-weight individuals but prevents leptin resistance and further the dysregulation of appetite and satiety in patients with morbid obesity. The decrease in BMI seems to be an effect exclusive to postmenopausal women. Therefore, further SNP analyses that include adjustments for levels of sexual steroids or female aging parameters, as well as the correlation of the SNP with these parameters, seem desirable in order to elucidate the sex-specific role of this SNP in obesity. In general, the highly sex-specific characteristics of leptin regulation should be the subject of further investigations. Although rs10487505 also influences obesity-related parameters in men, its effect on leptin and BMI seems to be found only in women. These results show that, at least regarding leptin, there should be a differential approach to understanding obesity in both sexes.

## Figures and Tables

**Figure 1 genes-14-00378-f001:**
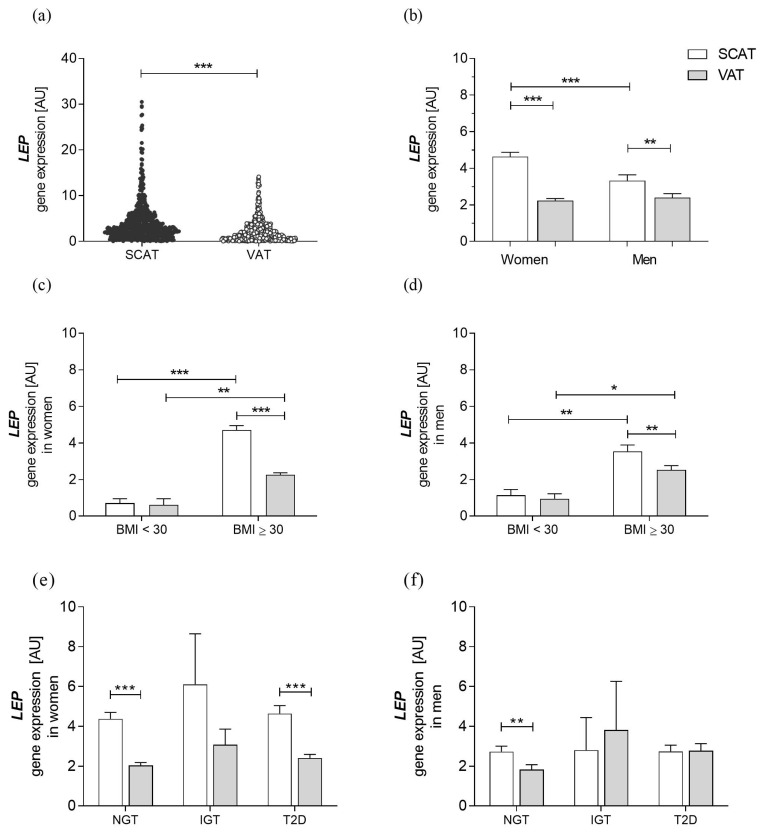
Characterization of *LEP* gene expression. *LEP* gene expression in (**a**) the entire cohort (N = 585) in subcutaneous (SCAT) and visceral (VAT) adipose tissue, (**b**) SCAT and VAT separated by sex (N = 429 in women, n = 156 in men), (**c**,**d**) obese patients (BMI ≥ 30 kg/m^2^; N = 420 in women, N = 143 in men) in comparison to the control group (normal weight and overweight individuals with BMI < 30 kg/m^2^; N = 6 in women, N = 14 in men) in SCAT and VAT respectively and (**e**) individuals with different degrees of glucose intolerance (for NGT N = 190 in women, N = 73 in men; for IGT N = 10 in women, N = 3 in men; for T2D N =157 in women, N = 68 in men). Group comparing analysis was performed on ln-transformed data by two-sided *t*-test or Welch’s test in case of unequal variances for (**a**–**d**) and by ANOVA with posthoc Bonferroni test for (**e**,**f**). Data are shown as mean ± SEM. * *p* < 0.05; ** *p* < 0.01; *** *p* < 0.001. NGT: normal glucose tolerance; IGT: impaired glucose tolerance; T2D: Type 2 Diabetes mellitus.

**Figure 2 genes-14-00378-f002:**
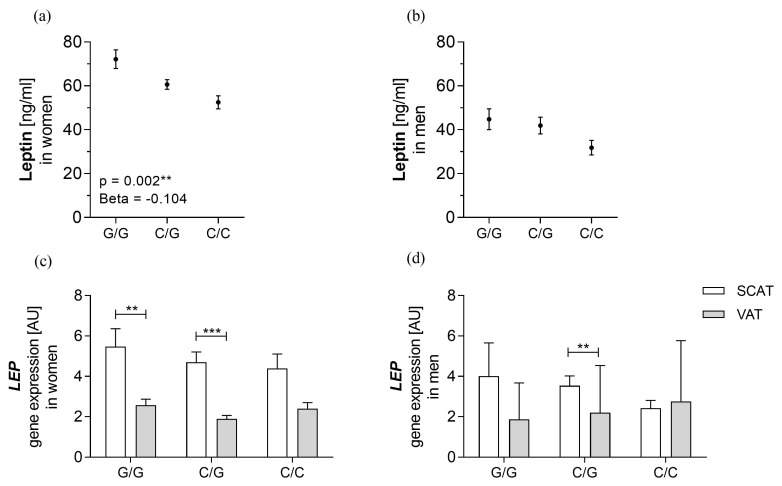
Association of rs10487505 with (**a**,**b**) circulating leptin and (**c**,**d**) *leptin* mRNA in both ATs. Data are shown as mean ± SEM. * *p* < 0.05; ** *p* < 0.01; *** *p* < 0.001 p adjusted for age and BMI.

**Figure 3 genes-14-00378-f003:**
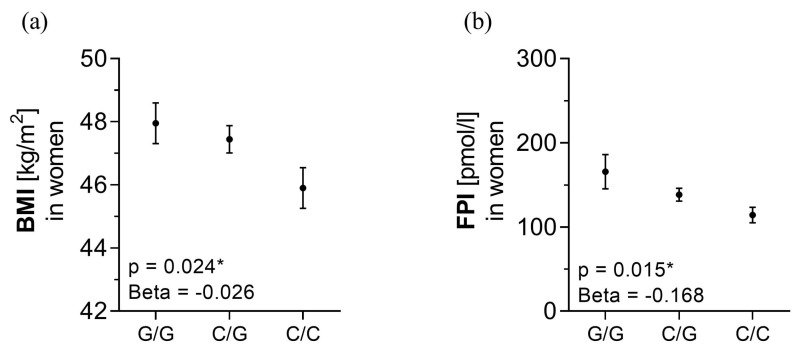
Association analysis of rs10487505 with (**a**) body mass index (BMI) and (**b**) fasting plasma insulin (FPI) in women. Data are shown as mean ± SEM. * *p* < 0.05. p adjusted for age (**a**,**b**) and BMI (only (**b**)).

**Table 1 genes-14-00378-t001:** Anthropometric and metabolic characteristics of the cohort studied.

	BMI < 30 kg/m^2^	BMI ≥ 30 kg/m^2^	*p*-Value
	(N = 138)	(N = 1497)	
Age (years)	61.77 ± 15.57	47.16 ± 12.03	**≤0.001**
Women/Men (N)	71/67	1042/455	
T2D (N)	23	622	
Body weight (kg)	69.74 ± 12.13	143.94 ± 31.13	**≤0.001**
Height (m)	1.69 ± 0.09	1.70 ± 0.10	0.194
BMI (kg/m^2^)	24.34 ± 3.19	49.18 ± 8.92	**≤0.001**
Waist circumference (cm)	84.33 ± 17.32	138.61 ± 21.05	**≤0.001**
Hip circumference (cm)	91.81 ± 13.52	145.41 ± 18.72	**≤0.001**
WHR	0.91 ± 0.11	0.96 ± 0.11	**0.001**
Body fat (%)	20.87 ± 4.50	47.12 ± 8.79	**≤0.001**
CRP (mg/l)	13.48 ± 25.87	11.35 ± 13.76	0.358
FPG (mmol/l)	5.85 ± 1.37	6.49 ± 2.67	**≤0.001**
FPI (pmol/l)	57.41 ± 77.03	148.58 ± 142.14	**≤0.001**
HbA1c (%)	5.60 ± 0.72	6.19 ± 1.31	**≤0.001**
Total cholesterol (mmol/l)	5.08 ± 1.12	4.74 ± 1.06	**0.036**
HDL-C (mmol/l)	1.64 ± 0.51	1.15 ± 0.44	**≤0.001**
LDL-C (mmol/l)	3.00 ± 1.07	3.03 ± 0.95	0.781
Triglycerides (mmol/l)	1.22 ± 0.52	2.26 ± 15.78	0.629
ALAT [µkat/l]	0.45 ± 0.30	0.77 ± 1.69	**0.028**
ASAT [µkat/l]	0.49 ± 0.26	0.81 ± 4.02	0.368
gGT [µkat/l]	1.03 ± 1.52	0.85 ± 2.83	0.472
Leptin serum levels [ng/mL]	8.25 ± 6.87	57.77 ± 35.42	**≤0.001**
IL-6 serum levels [pg/mL]	2.69 ± 3.24	6.74 ± 5.80	**≤0.001**
Adiponectin serum levels [μg/mL]	12.71 ± 6.19	6.54 ± 3.49	**≤0.001**
Subjects with *leptin* mRNA expression in AT (N)	4	306	

Data are given as means ± SD. Group comparison was performed by *t*-test or Welch’s test in case of unequal variances. Statistically significant difference between groups (*p* < 0.05) are highlighted in bold. AT: adipose tissue; BMI: body max index; WHR: waist-to-hip ratio; CRP: C-related protein; FPG: fasting plasma glucose; FPI: fasting plasma insulin; HbA1c: glycated hemoglobin; HDL-C: high-density lipoprotein cholesterol; LDL-C: low-density lipoprotein cholesterol; ALAT: alanine aminotransferase; ASAT: aspartate aminotransferase; gGT: γ-glutamyl transpeptidase; IL-6: interleukin 6; AT: adipose tissue; AU: arbitrary units; N: number.

**Table 2 genes-14-00378-t002:** Logistic regression between rs10487505 and occurrence of T2D in the additive model.

Genetic Variants	NGT N (%)	T2D N (%)	MAF NGT/T2D	*p*-Value (OR [95% CI]) Adj. for Age, Sex, BMI
rs10487505 (C/G)				
Entire cohort				
CC	186 (23.8%)	131 (20.2%)	0.4943/0.4645	0.433(1.069 [0.905–1.262])
CG	402 (51.3%)	340 (52.5%)		
GG	195 (24.9%)	177 (27.3%)		
Women				
CC	148 (26.2%)	77 (14.6%)	0.5133/0.3510	0.205(1.142 [0.930–1.403])
CG	283 (50.2%)	216 (41.0%)		
GG	133 (23.6%)	234 (44.4%)		

NGT: subjects with normal glucose tolerance; T2D: subjects with Type 2 Diabetes mellitus; MAF: minor allele frequency; CT: confidence interval; OR: odds ratio.

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
