# Peer review of "Sex-Specific Effects of the Genetic Variant rs10487505 Upstream of leptin in the Development of Obesity"

_genes, 2023, doi:10.3390/genes14020378_

Round 1
Reviewer 1 Report
In this manuscript by Molder et al., the authors identified a genetic variant rs10487505 SNP upstream of leptin associated with obesity in humans. Although the potential association between leptin SNP variants and obesity in women is interesting, the sex-specific studies are rather weak to support the conclusions of this manuscript. The authors explained the SNP may not or not exclusively regulate leptin gene expression. Perhaps, I understand the difficulty of providing additional data in such a short fixed time. This manuscript could be improved with discussion insights on why the leptin-lowering effect of the C allele is in a sex-dependent or postmenopausal-dependent manner.
Minor comment
1. P-value indications for significance are missing for several Figure graphs (Figure 1e-NGT/T2D, Figure 1f-NGT, and Figure 2c).
Author Response
"Please see the attachment."

Reviewer 2 Report
The hypothesis of this manuscript and the methods used to assess the lectin-lowering effect of the rs10487505 variant in European men and women adjusted for BMI, sex, and age to elucidate the influence of rs10487505 on leptin mRNA expression are well conducted. The results and discussion are consistent with previous studies and allow the generation of new hypotheses to better understand the influence of the genetic variant (rs10487505) on leptin mRNA expression.
However, to improve this manuscript the authors should attend to the following suggestion:
The genetic variant rs10487505 is a common variant in humans, however the frequency of the C allele is quite different between various populations, for example, in Europeans the frequency is 0.47 in Asians (and East Asians) it is 0.75 and in Latin Americans (non-Europeans) is 0.39, on the other hand, the BMI is one of the anthropometric traits with large differences between large populations. The sample used in this work included a European cohort and the results and discussions are based on the European population adjusted for sex, age and BMI, the suggestion for this manuscript is to write a perspective paragraph on the influence of ancestry on this phenotype of leptin in the development of obesity in various populations? I'm not sure this suggestion is difficult to figure out and write, but I think it's important to comment on the influence of ancestry in this leptin study. Do you think there may be some influence of ancestry on your results for this phenotype?
Some useful references to help write this paragraph:
1. Project GIANT (multi-ethnic version) for BMI and gender specific
https://portals.broadinstitute.org/collaboration/giant/index.php/GIANT_consortium_data_files#2018_Exome_Array_Summary_Statistics
2. Global Lipid Genetics Consortium (multiple ancestry meta-analysis)
http://csg.sph.umich.edu/willer/public/glgc-lipids2021/
3. https://t2d.hugeamp.org/variant.html?variant=rs10487505
4. https://www.ncbi.nlm.nih.gov/snp/rs10487505
Author Response
"Please see the attachment."

Reviewer 3 Report
In this study, the authors investigated the impact of rs10487505 on BMI, leptin mRNA expression in AT, circulating leptin levels, and obesity-related parameters stratified by gender. In my opinion, the study is interesting and well-designed. I have several concerns that need to be addressed, as follows:
1- Materials and Methods:
- The circulating leptin was measured. However, this issue was not specified in the methods. Please define how circulating leptin is being measured.
- 2. Leptin mRNA expression analyses in AT: Please specify the method of mRNA expression calculation.
- Please define any additional data gathered for obesity-related parameters.
2- Discussion: There are several limitations that should be acknowledged; for example, the pathogenic mechanisms for the impact of rs10487505 on leptin mRNA expression were not investigated, etc... Please define the limitations at the end of this section.
3- The conclusion section is unnecessarily long and should summarize the main study findings.
Author Response
"Please see the attachment."
